# The Effects of National Fundamental Factors on Regional House Prices: A Factor-Augmented VAR Analysis

**Xiang Gao \*, Wen Kong \* and Zhijun Hu**

School of Finance, Jiangxi University of Finance and Economics, No.168, East Shuanggang Road, Nanchang 330013, China; hzhj.0325@163.com
\* Correspondence: gaoxiang3723@foxmail.com (X.G.); kwenedu@163.com (W.K.)

**Abstract:** Using panel data from 30 regions in China during the period 1999:01–2020:12, this paper evaluates the effects of national fundamentals affecting the movement of regional house prices by estimating a factor-augmented VAR model. We construct and examine a hypothesis that national fundamentals affecting regional house prices, such as monetary policy (short-term interest rate and M2), real output, and inflation rate, may affect regional house prices through their impacts on common factors. The empirical results show that monetary shocks (both interest rate and M2) can significantly affect regional house prices, but the effects are pretty different across regions. However, the effects of the real output and inflation rate are less important. Therefore, this study offers valuable information for regulators to improve the effectiveness of monetary policy to stabilize house markets from a regional perspective.

**Keywords:** national fundamental factors; regional house price; monetary policy; factor-augmented VAR model

## 1. Introduction

The development of house markets has attracted much research interest over the past decades, as the movement of house markets is identified as an important factor affecting household welfare and economic activities (Iacoviello 2005; Iacoviello and Minetti 2008; Mian et al. 2013; etc.). Economists typically investigate the relationship between macroeconomic variables and house prices based on the aggregate indicators relating to houses. However, the empirical analysis may be misleading from the perspective of national house prices, as this type of practice does not take account of the heterogeneity and local characteristics across regions in house markets.

Since the beginning of the 2000s, the rapid growth of house prices has been a remarkable phenomenon in China. The unprecedented house market boom has attracted the attention of authorities and economic analysts. A large number of studies have documented "housing bubbles" and the relationship between house markets and macroeconomic variables from a national perspective (Li and Chiang 2012; Wang and Zhang 2014; Xu et al. 2016; Chen and Wen 2017; etc.). However, there is little literature analyzing the relationship between macroeconomic variables and house markets from a regional perspective. Figure 1 provides a picture of co-movement among house prices as well as the heterogeneity in the growth rates of house price across regions in China. For example, the annual growth rates of house prices are higher than 11% in Shanghai and Hainan, whereas the growth rates are just slightly higher than 4% in Yunnan and Xinjiang. The average annual growth rates of house prices are higher than 8% in only seven regions (See Table 1). Thus, the data imply that the house price boom may not be a national-wide phenomenon and that policy evaluation based on the national average of house prices may be misleading.

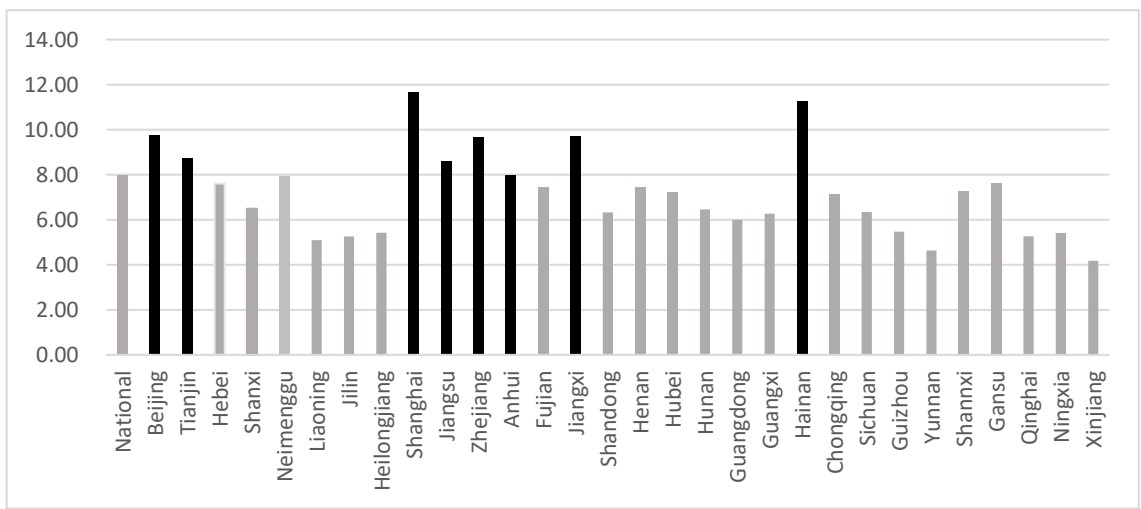

**Figure 1.** Average annual growth rates of regional house prices. Note: The vertical axis represents the house price growth rate (%), the horizontal axis represents the country and 30 regions. The regions that are labeled black are the regions with higher growth rates than national. All growth rates are calculated by the authors.

**Table 1.** Average annual growth rates of house prices for 30 regions in China.

| Order | Region | Growth (%) | Order | Region | Growth (%) |
|-------|--------|-----------|-------|--------|-----------|
| 1 | Shanghai | 11.64 | 17 | Chongqing | 7.14 |
| 2 | Hainan | 11.25 | 18 | Shanxi | 6.49 |
| 3 | Beijing | 9.74 | 19 | Hunan | 6.46 |
| 4 | Jiangxi | 9.70 | 20 | Sichuan | 6.35 |
| 5 | Zhejiang | 9.64 | 21 | Shandong | 6.32 |
| 6 | Tianjin | 8.72 | 22 | Guangxi | 6.27 |
| 7 | Jiangsu | 8.59 | 23 | Guangdong | 6.00 |
| 8 | National | 7.99 | 24 | Guizhou | 5.47 |
| 9 | Anhui | 7.98 | 25 | Heilongjiang | 5.42 |
| 10 | Neimenggu | 7.95 | 26 | Ningxia | 5.42 |
| 11 | Gansu | 7.64 | 27 | Qinghai | 5.27 |
| 12 | Hebei | 7.61 | 28 | Jilin | 5.26 |
| 13 | Fujian | 7.44 | 29 | Liaoning | 5.10 |
| 14 | Henan | 7.42 | 30 | Yunnan | 4.64 |
| 15 | Shaanxi | 7.27 | 31 | Xinjiang | 4.18 |
| 16 | Hubei | 7.24 | - | - | - |

Note: The table shows the average annual growth rates of national house prices and 30 regions' house prices, covering the period 2000M01–2020M12. All growth rates are calculated by the authors.

Motivated by the data, this paper aims to evaluate the effects of national fundamentals such as monetary policy, real output, and inflation rate on the co-movement of regional house prices through common factors. For this, we utilize a large monthly dataset (155 variables) from 30 regions in China and estimate a factor-augmented VAR model (FAVAR) proposed by Bernanke et al. (2005). Thus, we can construct and examine a hypothesis that national fundamentals may affect regional house prices through their impacts on common factors. The FAVAR model works efficiently in a large information set by extracting a small number of latent factors from the large pool of observed data series. Thus, it allows for the use of multiple indicators of economic concepts without assuming that the economic concepts are observed. In the current model, the cross-region heterogeneity is reflected in a way that national fundamentals affect common factors, and the changes in common factors in turn affect regional house prices through the factor loading coefficient of each region.

Our study is related to the literature that examines the relationship between national fundamentals and house prices. However, we contribute to the literature in several aspects.

Firstly, differently from others, as it examines the relationship from a regional perspective, we could identify the effects of national fundamentals on 30 regions' house prices by 30 impulse responses from a single FAVAR system with 155 variables. The high dimensional system may yield more statistically significant results. Secondly, the empirical results show that monetary shocks can significantly affect regional house prices through common factors, but the effects are pretty different across regions, which means the monetary shocks can significantly affect regions with high growth rates instead of low growth rates. Thirdly, differently from others, we find that the effects of the national real output and inflation rate are less important to the 30 regions' house price movements. Our analysis offers valuable information for regulators to improve the effectiveness of monetary policy to stabilize house markets from a regional perspective.

The remainder of the paper is structured as follows: Section 2 presents the literature review. Section 3 introduces data, and Section 4 describes methodology. Section 5 provides empirical results. Finally, Section 6 concludes the study.

## 2. Literature Review

A large number of studies have investigated the relationship between national fundamental factors and regional house prices in industrial countries since the beginning of the 1990s (e.g., Hwang and Quigley 2006; Glindro et al. 2011; Gu 2018; etc.). Otrok and Terrones (2005) estimate a FAVAR model to evaluate the linkages among international house prices, interest rates, and real activity, and they find that the movement in house prices can significantly affect macroeconomic aggregates. Furthermore, the monetary shocks can significantly affect the house price fluctuations both in the U.S. and internationally. In the U.S., Negro and Otrok (2007) use a dynamic factor model estimated via Bayesian methods to investigate the co-movement of house prices in the U.S. They identify three components in the movement of house prices in the model: national, regional, and state-specific components. They find that the movement in house prices is mainly driven by the specific regional factors (e.g., Stock and Watson 2005, 2011; Kallberg et al. 2014; etc.). In the Eurozone economies, Merikas et al. (2012) use a cointegration approach and a VAR system to examine house price co-movement They provide strong evidence of the importance of local factors (especially the interest rate) on the co-movement of house prices. Taking into account the substantial impacts of house price movements on mortgage values, substantial research has focused on the role of house prices in both the amplification and the propagation of shocks to economic activities (e.g., Iacoviello 2005; Hwang and Quigley 2006; Goodhart and Hofmann 2008; etc.).

Dramatic growth in China's house market has attracted growing research attention since the beginning of 2000s. A large number of studies have investigated the housing bubble in China (e.g., Dreger and Zhang 2013; Chen and Wen 2017; etc.). A few works of literature document the co-movement of house prices, with the common feature mainly focusing on the ripple effect. Weng and Gong (2017) apply a DSP-GJR-GARCH model to investigate price co-movement and volatility spillover effects in China's housing markets over the period 2005~2014. They find that the co-movement of house prices is significantly affected by the population, income, and national macroeconomic situation (e.g., Zhang et al. 2012; Chiang 2010; etc.).

There is little literature investigating the relationship between national fundamentals and house markets; most of the literature examines the effects of national fundamentals on house markets from the perspective of economic zones (e.g., Wei and Wang 2010, etc.). Liang and Gao (2007) use an error correction model and a panel data model to analyze the effect of monetary policy on the fluctuations of real estate price in China. They estimate the models for three groups (eastern, central, and western groups) and find that the effect of monetary policy is effective in the eastern group and western group but not in the central group (e.g., Wei and Wang 2010, etc.). Yu (2010) estimates a dynamic panel data model and finds that there is no stable relationship between house prices and economic fundamentals in China (e.g., Li and Chiang 2012; Xu et al. 2016).

## 3. Data

The dataset in this paper consists of regional datasets from 30 regions and a national dataset covering the period from the first month of 1999 to the last month of 2020. The regional monthly datasets consist of house prices, the industry added value, loans to real estate corporations, the consumer consumption index, and exports. We obtain the regional house price indicators by dividing the total sales volume by the total sales area. Tables 1 and 2 show the average annual growth rates and summary statistics of national and regional house prices. The heterogeneous growth rates in the house prices across regions could be caused by national factors and regional factors. In order to cover the regional factors affecting the regional house prices, we choose the other four regional factors as specific regional factors. Thus, we get 150 ($5 \times 30 = 150$) regional variables. The high dimensional variables enable us to extract many more efficient common factors in the FAVAR system.

**Table 2.** Summary statistics of national and regional house prices.

|  | Obs | Mean | Std. Dev. | Min | Max |
|---|---|---|---|---|---|
| National | 252 | 3362.54 | 1062.87 | 1835.57 | 5435.28 |
| Beijing | 252 | 9723.48 | 5321.73 | 2698.28 | 23,247.34 |
| Tianjin | 252 | 4774.88 | 2119.66 | 1755.34 | 9639.42 |
| Hebei | 252 | 2400.84 | 971.12 | 991.80 | 4578.61 |
| Shanxi | 252 | 2026.91 | 804.49 | 811.77 | 3578.02 |
| Neimenggu | 252 | 1891.24 | 806.14 | 721.22 | 3252.92 |
| Liaoning | 252 | 3140.34 | 750.95 | 1496.37 | 4650.05 |
| Jilin | 252 | 2600.58 | 859.25 | 998.32 | 3984.60 |
| Heilongjiang | 252 | 2536.08 | 849.22 | 1429.78 | 4107.46 |
| Shanghai | 252 | 8048.22 | 4306.23 | 2066.87 | 18,696.78 |
| Jiangsu | 252 | 3463.54 | 1445.87 | 1385.82 | 6388.06 |
| Zhejiang | 252 | 5106.49 | 2725.78 | 1516.35 | 8970.48 |
| Anhui | 252 | 2407.20 | 1072.60 | 764.26 | 4130.95 |
| Fujian | 252 | 4098.95 | 1908.78 | 1629.91 | 7487.22 |
| Jiangxi | 252 | 2051.76 | 1153.49 | 663.53 | 4164.09 |
| Shandong | 252 | 2596.71 | 919.29 | 1331.48 | 4200.43 |
| Henan | 252 | 2020.14 | 737.93 | 622.26 | 3617.72 |
| Hubei | 252 | 2583.16 | 994.22 | 1198.35 | 4729.60 |
| Hunan | 252 | 1899.34 | 770.93 | 682.63 | 3147.67 |
| Guangdong | 252 | 4942.30 | 1645.54 | 2552.75 | 8580.75 |
| Guangxi | 252 | 2297.88 | 723.62 | 1170.22 | 3689.77 |
| Hainan | 252 | 4561.81 | 2491.89 | 893.20 | 8586.81 |
| Chongqing | 252 | 2529.28 | 1136.32 | 924.48 | 4292.71 |
| Sichuan | 252 | 2384.66 | 1014.75 | 974.71 | 3850.98 |
| Guizhou | 252 | 1854.02 | 713.03 | 894.43 | 3071.25 |
| Yunnan | 252 | 2280.36 | 617.93 | 1100.15 | 3698.02 |
| Shannxi | 252 | 2498.03 | 960.79 | 843.86 | 3976.28 |
| Gansu | 252 | 1951.29 | 767.03 | 757.09 | 3807.09 |
| Qinghai | 252 | 1872.00 | 583.77 | 929.30 | 3128.89 |
| Ningxia | 252 | 2011.21 | 567.11 | 913.37 | 3047.50 |
| Xinjiang | 252 | 1984.37 | 633.48 | 1066.01 | 3325.86 |

Note: The unit is Yuan/m$^2$. For the summary statistics of the rest, variables are available, if required.

The national monthly dataset consists of the national house price (NHP), industrial production index (IP), consumer price index (P), short-term interest rate (one-night call rate, R), and M2. Thus, we could investigate the effects of the five national fundamental variables on the 30 regions' house prices through the common factors extracted from the 150 regional variables.

We obtained both the national and regional monthly datasets from the Wind database. This paper will be working with the growth rate of each variable, except for the interest rate. The current paper uses both the short-term interest rate and M2 as the monetary policy instruments. There are two main reasons to use both policy instruments: (1) Attributed

to the undergoing reform policy, the role of interest rates has been much more significant since the beginning of the 2000s; substantial literature (e.g., Liao and Tapsoba 2014; Kamber and Mohanty 2018; etc.) has provided strong evidence of the significant effect of interest rates as a monetary policy instrument. (2) In the spirit of Sims (1992), the monetary policy shocks (interest rate) could lead to the problem of a "price puzzle" due to the effect of market expectations, but not M2. Thus, we can find out the difference of the effects between the interest rate and M2.

Table 3 presents the descriptions of variables and Table 4 presents the summary statistics of the variables used in this paper. Considering so many variables (155 variables) in this study, we only present the summary statistics of the national varibles and Beijing Province; for the rest, 145 variables are avaiable, if required.

**Table 3.** Variable descriptions.

|  | Variables | Abbreviation | Time Series |
|---|---|---|---|
| National | house price | NHP | 1999:01–2020:12 |
|  | industrial production index | IP | 1999:01–2020:12 |
|  | consumer price index | P | 1999:01–2020:12 |
|  | short-term interest rate | R | 1999:01–2020:12 |
|  | monetary supply | M2 | 1999:01–2020:12 |
| Regional | house price | HP | 1999:01–2020:12 |
|  | industry added value | IAV | 1999:01–2020:12 |
|  | loans to real estate corporations | LOAN | 1999:01–2020:12 |
|  | consumer consumption index | CPI | 1999:01–2020:12 |
|  | export | EX | 1999:01–2020:12 |

**Table 4.** Summary statistics of the growth rates of variables (part of variables).

|  | Variable | Obs | Mean | Std. Dev. | Min | Max |
|---|---|---|---|---|---|---|
| National | NHP | 252 | 0.41 | 3.48 | −13.61 | 13.05 |
|  | IP | 252 | 9.41 | 4.27 | −3.26 | 23.54 |
|  | P | 252 | 2.19 | 2.06 | −1.81 | 8.34 |
|  | R | 252 | 2.22 | 0.72 | 0.81 | 6.43 |
|  | M2 | 252 | 15.20 | 3.25 | 9.14 | 25.96 |
| Beijing | HP | 252 | 9.70 | 17.43 | −45.21 | 50.42 |
|  | IAV | 252 | 1.81 | 2.70 | −4.74 | 10.46 |
|  | LOAN | 252 | 12.98 | 30.91 | −84.84 | 101.95 |
|  | CPI | 252 | 6.36 | 18.40 | −41.47 | 69.11 |
|  | EX | 252 | 7.62 | 7.40 | −15.49 | 26.79 |

Note: for the summary statistics of the rest, 145 variables are available, if required. All growth rates are calculated by the authors.

In consideration of the Chinese New Year holiday's effect on monthly variables, we perform the same smooth process to calculate the missing data[1]. This paper deflates the house prices and economic activities by inflation rates. In consideration of seasonal effects, all economic activities and prices are seasonally adjusted using the Census X-12 ARIMA method.

## 4. Methodology

Since its advent decades ago, vector autoregressive (VAR) methods have been widely used in empirical economic analyses. Numerous works of literature (e.g., Bernanke and Blinder 1992; Sims 1992; etc.) have used VAR methods to examine the effects of monetary policy shocks on macroeconomic variables and prices. Even though the standard VAR system is useful in measuring the effects of monetary policy shocks and providing an empirical forecast, there are some considerable limitations. The most significant limitation is the problem of degree-of-freedom. Bernanke et al. (2005) points out that the limited information set used in the VAR model can lead to at least three problems: first, the VAR model cannot include all of the necessary information sets, which can lead to a biased measurement of the monetary policy shock; second, the choice of a specific observable to represent an economic concept is often arbitrary to some degree; third, the sparse information set can lead to limited impulse response functions for the limited variables.

In order to solve these problems arising in the standard VAR model, this paper uses a FAVAR model proposed by Bernanke et al. (2005). The FAVAR model works efficiently in a large information set by extracting a small number of latent factors from a large pool of observed data series. Thus, it allows for the use of multiple indicators of economic concepts without assuming that the economic concepts are observed. Then, the FAVAR model can directly determine whether the effect of additional information is significant or not. In a dynamic factor model (Stock and Watson 1989, 2002; Bai et al. 2016; etc.), a small number of essential common factors can be extracted from a broad data set—for example, a common factor can be extracted from the regional data series on house prices, the industry added value, loans to real estate corporations, the consumer consumption index, and exports in the current paper. By focusing attention on the essential common factor, the dimensionality of the model can be greatly reduced, allowing for the estimation of a FAVAR model.

In particular, we assume that the regional data series, $X_t$, are determined by a vector of observables, $Y_t$, a small number of underlying factors, $F_t$, and idiosyncratic noise, $e_t$, according to:

$$X_t = \wedge^f F_t + \wedge^y Y_t + e_t \tag{1}$$

where $X_t$ is an N × 1 vector of the observable informational dataset, and N >> M + K. The size of the vector $F_t$ is less than the that of the vector of $X_t$. $Y_t$ and $e_t$ are N × 1 vectors of the error term with a mean of zero and a finite variance W. The cross-correlation in $e_t$ dies out as N goes to infinity. $\wedge^y$ is an N × M matrix of factor loadings, and $\wedge^f$ is an N × K matrix of factor loadings.

In our study, $X_t$ is specified as the vector of 30 regions' house prices, the industry added value, loans to real estate corporations, the consumer consumption index, and exports; $F_t$ is specified as the vector of common factors extracted from the 150 regional variables. Examining the exact numbers of factors is not the main issue of this study; the impulse responses with different numbers of common factors will be placed in the Appendix A section. $Y_t$ is specified as the vector of the national fundamental variables (IP, P, R, M2).

In a dynamic factor model, the factors F are related over time, typically according to the linear process:

$$F_t = A(L)F_{t-1} + \epsilon_t \tag{2}$$

where $A(L)$ denotes a polynomial in the lag operator, and $\epsilon_t$ follows the standard normal distribution.

In the spirit of Bernanke et al. (2005), a FAVAR model is a VAR in which a small number of common factors are taken from a dynamic factor model:

$$\begin{bmatrix} F_t \\ Y_t \end{bmatrix} = \varnothing(L) \begin{bmatrix} F_{t-1} \\ Y_{t-1} \end{bmatrix} + \eta_t \tag{3}$$

In this transition equation, $Y_t$ is an M $\times$ 1 vector of observable economic variables. $F_t$ is a K $\times$ 1 vector of unobservable factors. $\varnothing(L)$ is a polynomial with lag $L$. The terms of $\varnothing(L)$ that related $Y_t$ to $F_{t-1}$ should not be zero; otherwise, it turns into a standard VAR model without any latent factors. $\eta_t$ is an error term with a mean of zero and a finite covariance matrix $Q$. If the true model is a FAVAR, then the estimation of Equation (3) as a standard VAR model in $Y_t$, with the factors omitted, will generally lead to biased estimates of the VAR coefficients and related quantities of interest, such as impulse response coefficients.

Note that Equation (1) cannot be estimated directly because the latent factors $F_t$ are unobservable; therefore, principal component methods are used to estimate $F_t$ in the dynamic factor model. We utilize a two-step principal components approach, which provides a nonparametric way of uncovering the space spanned by the common components. Based on Stock and Watson (1989, 2002, etc.), in the first step, the common factors, $Z_t$, are estimated using the first K + M principal components of $X_t$. Note that the estimation of the first step does not exploit the fact that $Y_t$ is observed.

$$\hat{Z}_t = b_F \hat{F}_t + b_Y Y_t + \zeta_t \tag{4}$$

$\hat{F}_t$ is obtained as the part of the space covered by $\hat{Z}_t$ that is not covered by $Y_t$. So, in the first step, the common factors are obtained entirely from Equation (1), and the identification of the factors is standard. In the second step, Equation (3) is estimated by standard methods, with $F_t$ replaced by $\hat{F}_t$. To obtain accurate confidence intervals on the impulse response functions reported below, we implement a bootstrap procedure, based on Kilian (1998), that accounts for the uncertainty in the factor estimation.

An important question is how we identify the shocks of fundamental factors. For Equation (3), we take the four fundamental factors as observables, i.e., the elements of Y. When we examine the shock of monetary policy (R, M2), we then assume a typical recursive ordering, with the real output and inflation rate first and then the policy variables. The recursive identification assumes that monetary policy can respond endogenously to changes in real output or inflation within the month but that policy innovations affect real output and inflation only with a lag. Similarly, we would examine the shocks of real output and inflation by the same process.

## 5. Empirical Results

### 5.1. Effects of Monetary Shocks

We estimate the FAVAR system with four-month lags for each of the variables, in accordance with the AIC and BIC lag-selection criteria. Figure 2 reports the impulse responses of regional house prices to positive shocks to interest rates, as obtained from the two-step FAVAR system with one factor. According to the figure, the contractionary monetary policy has economically and statistically significant effects on the house prices of 10 regions: Beijing, Shanghai, Tianjin, Zhejiang, Guangdong, Jiangsu, Anhui, Chongqing, Guizhou, and Ningxia. These regions, except for two regions (Guizhou and Ningxia), are identified as developed regions in China, with relatively higher per capita income levels as well as high house price growth rates. This result indicates that the interest rate instrument could affect the house prices in relatively developed regions in China. It is worthwhile to note that the interest rate shock could significantly affect the house prices of Guizhou and Ningxia, even though they are less developed regions with low per capita income levels and low growth rates of house prices. The current methodology is not an effective methodology to examine the reason, but it may be attributed to the policy called the Regional Coordinative Development Strategy[2]. This deserves further study with a more effective methodology.

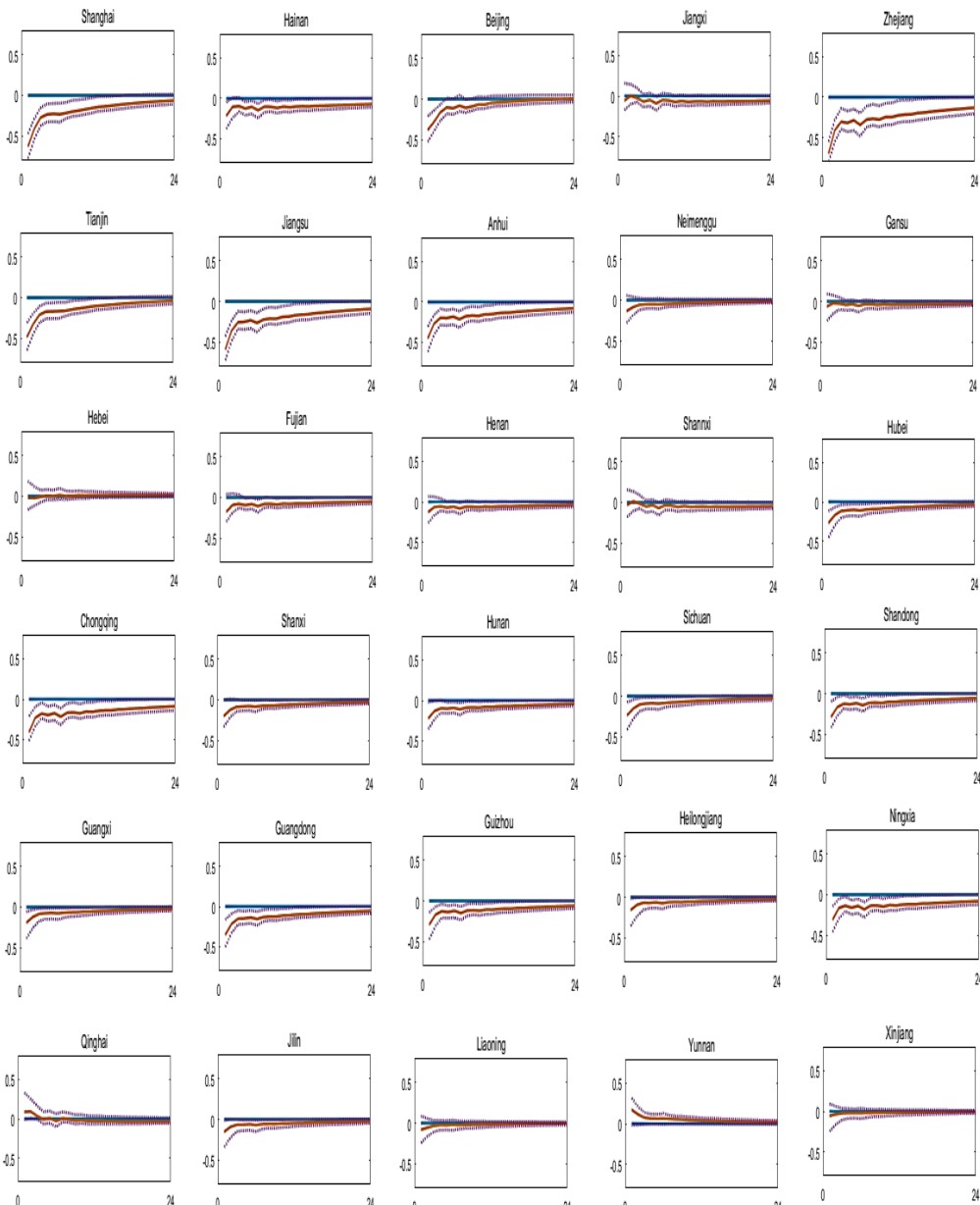

**Figure 2.** Impulse responses of regional house prices to one standard error shock to the interest rate: FAVAR system with one factor.

In the case of Hainan and Jiangxi, the impulse responses are not statistically significant in the 90 percent confidence intervals, even though Hainan and Jiangxi have higher growth rates but lower per capita income levels. The insignificant effect of interest rates suggests that the house price movements of Hainan and Jiangxi are not mainly impacted by national fundamental factors such as interest rates.

In sum, we find that the effects of interest rate shocks on the movement of house prices are quite different across regions. The effect is very strong in developed regions (Shanghai, Zhejiang, Jiangsu, etc.) but very weak in less developed regions (Liaoning, Gansu, etc.). This confirms our inference that the policy analysis based on the aggregate indicators may be misleading. The result suggests that the policymakers can implement contractionary monetary policies to curb the overheating house prices in the main regions.

Examining the exact number of factors is not the main issue of this study, and our focus is to test the sensitivity of the results to the alternative number of factors. Figure A1 presents the impulse responses of regional house prices to contractionary monetary policy

with three factors. The result suggests that the qualitative results on the effects of interest rate shocks are not altered by increasing the number of factors.

In Figure 3, we use an alternative measure of monetary policy stance, M2. The figure shows that the M2 shocks have economically and statistically significant effects on 10 regions: Zhejiang, Fujian, Hunan, Chongqing, Sichuan, Guizhou, Guangxi, Heilongjiang, Xinjiang, and Ningxia. Comparing Figures 2 and 3, we find that M2 shocks cannot significantly affect the house prices of the main developed regions, especially the three leading regions, Beijing, Shanghai, and Guangdong. On the other hand, the M2 shocks are more likely to affect the house prices of the regions with close geographical and economic proximities: Hunan, Jiangxi, Chongqing, Sichuan, Guizhou, and Guangxi are neighboring regions with close economic proximities.

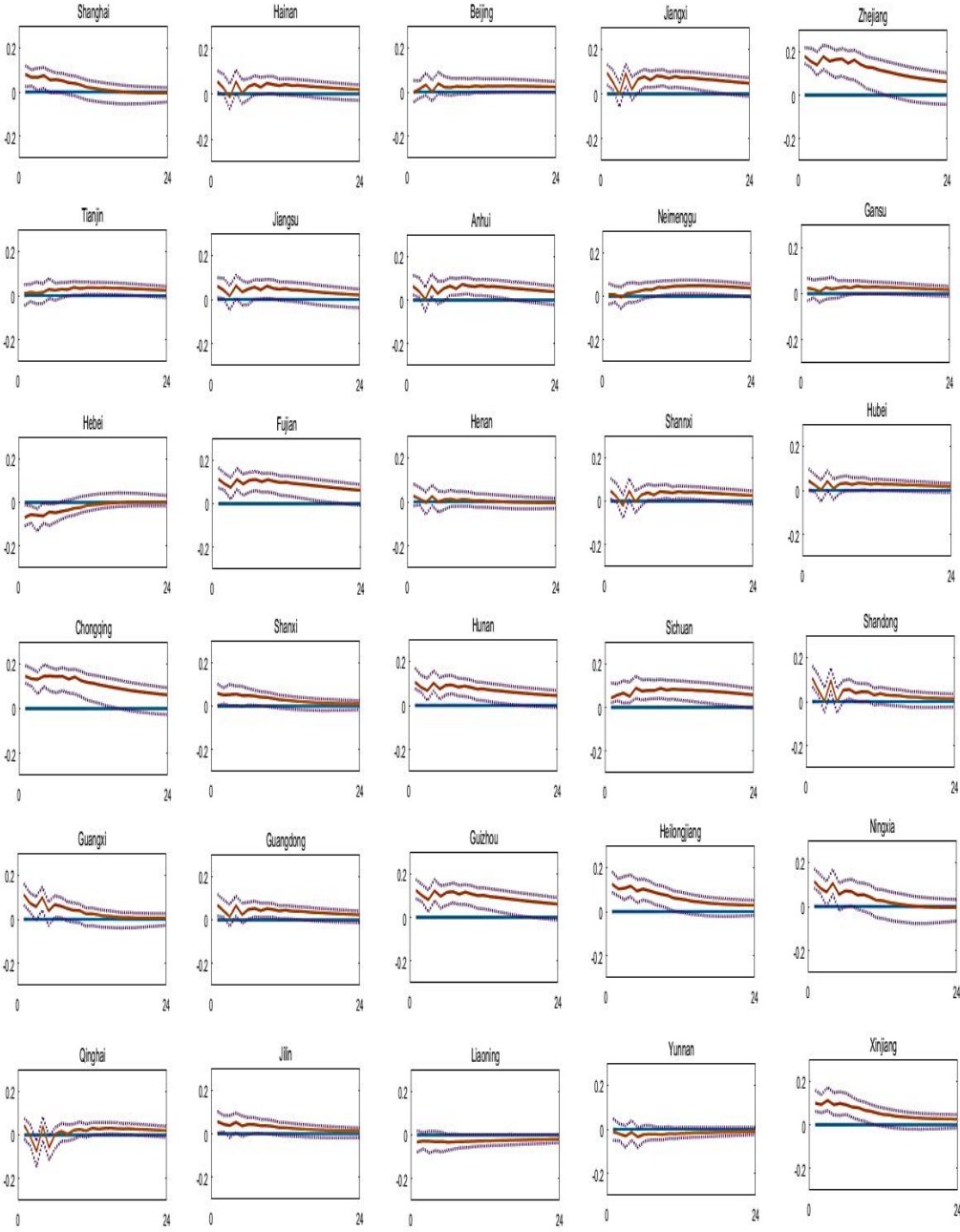

**Figure 3.** Impulse response of house prices to one standard error shock to M2: FAVAR system with one factor.

We also test the sensitivity of the results to the alternative number of factors. Figure A2 presents the impulse responses of regional house prices to M2 shocks with three factors; the impulse responses suggest that the qualitative conclusions on the effect of M2 are not altered by the use of three factors.

Figures 2 and 3 present the effects of monetary shocks (R, M2) on the movement of regional house prices. Our impulse response analysis shows two important findings. First, the monetary shocks can significantly affect regional house prices, but differently across regions. This confirms the inference that the analysis based on the national average of house prices may be misleading. Second, the effects of the interest rate and M2 are significantly different. The significantly different effects of the interest rate and M2 may be attributed to several reasons. First, the effect of interest rate shocks on the market expectation. Based on Sims (1992), this would lead to the problem of a "price puzzle", when we use the call rate as a monetary policy instrument. For instance, an expansionary monetary shock leads to higher inflation and people's expected inflation, but the higher people's expected inflation cannot be observed in the VAR system. Then, the policymakers design contractionary policy to curb the increasing inflation. Contractionary monetary policy leads to higher inflation due to the unobserved market expectation, which refers to the "price puzzle". Unlike the call rate, M2 as a monetary policy instrument can avoid the problem of market expectation. Second, the different developing levels of financial markets across regions. Following the financial market reform in China, the development of financial markets is quite heterogeneous across regions. The financial markets in the regions located in the east and south of China are much more developed than those located in the center and west of China. Third, differently from developed countries adopting interest rate targeting systems, China implements monetary policies by adopting monetary aggregate targeting systems.

*5.2. Effects of Real Output Shocks*

Figure 4 shows the impulse responses of regional house prices to one standard error shock to the real output, as obtained from the two-step FAVAR system with one factor. According to the figure, the positive shock to the real output has significant effects on the house prices in Zhejiang, Fujian, Jiangsu, Anhui, Sichuan, and Guizhou, but the effects are very short and weak. Note that these regions, except for Guizhou, are developed regions with higher per capita income levels, whereas the effects of real output shocks cannot significantly affect the house prices of the other regions in the 90 percent confidence intervals. Then, we also tested the sensitivity of the results to the alternative number of factors. Figure A3 presents the impulse responses of regional house prices to real output shocks with three factors, and the results suggest that the qualitative conclusions on the effect of the real output are not altered by the use of three factors.

Our impulse analysis shows that the impacts of the real output on the movement of regional house prices are very short and weak. The results imply that the movements of regional house prices may not be mainly affected by the national real output.

*5.3. Effects of Inflation Shocks*

Figure 5 shows the impulse responses of regional house prices to one standard error shock to the national inflation rate, as obtained from the two-step FAVAR system with one factor. According to the figure, the impulse responses are not statistically significant in the 90 percent confidence intervals for all regions. Then, we also tested the sensitivity of the results to the alternative number of factors. Figure A4 presents the impulse responses of regional house prices to national inflation rate shocks with three factors; the result shows that the qualitative conclusions on the effect of the inflation rate are not altered by increasing the number of factors. This implies that the movement of regional house prices may not be affected by the national inflation rate.

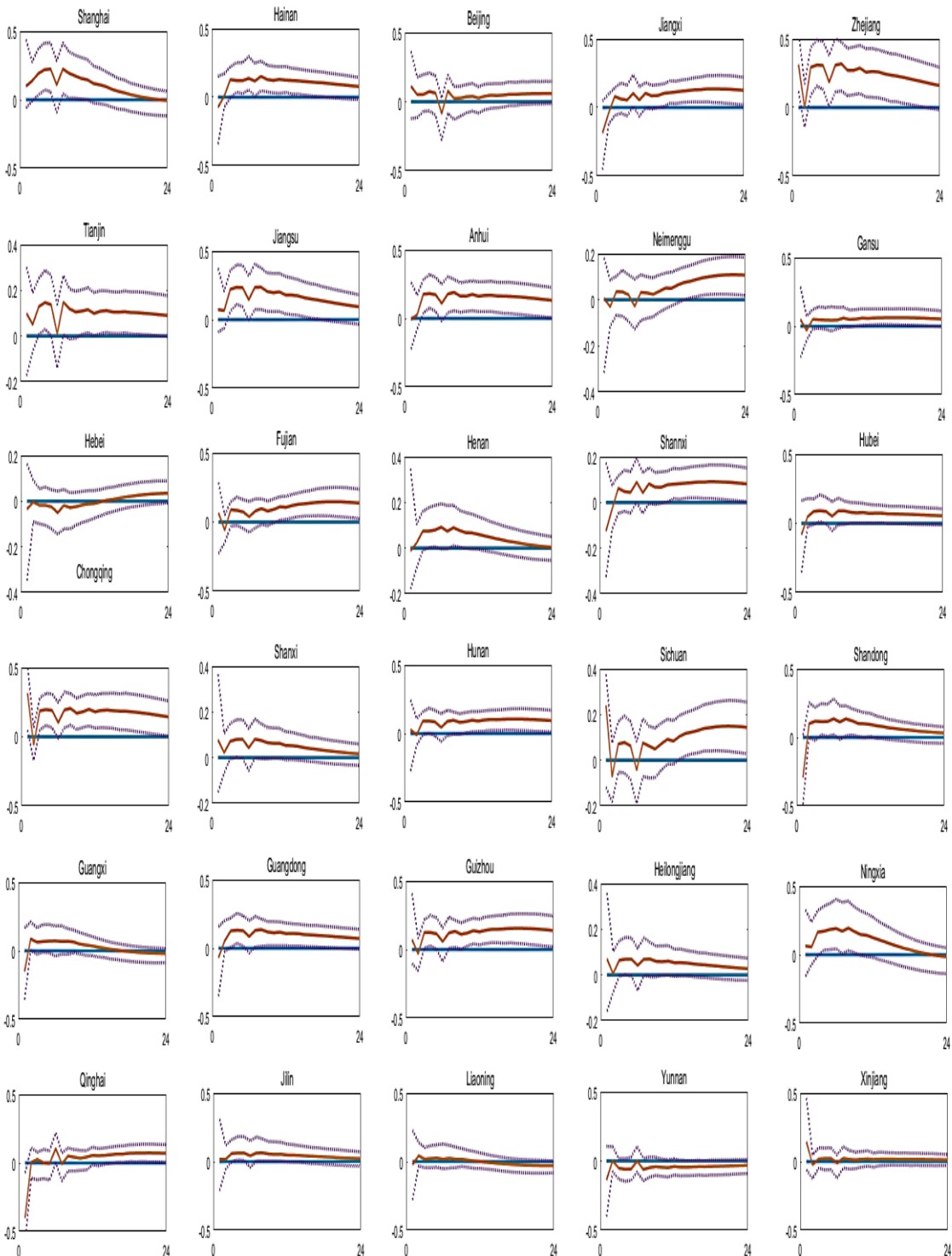

**Figure 4.** Impulse responses of regional house prices to one standard error shock to real output: FAVAR system with one factor.

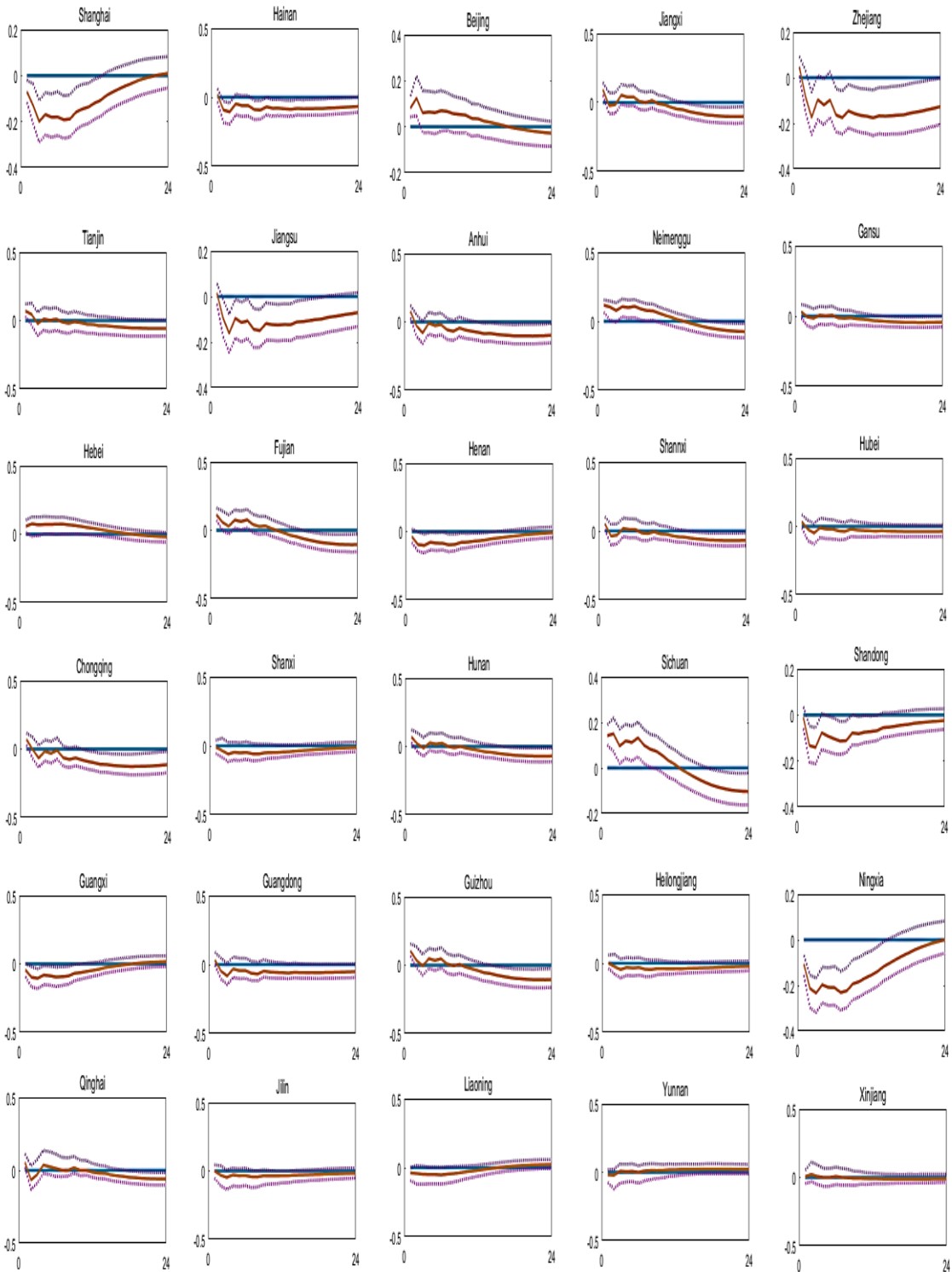

**Figure 5.** Impulse responses of regional house prices to one standard error shock to inflation rate: FAVAR system with one factor.

## 6. Conclusions

Motivated by the heterogeneity of the growth rates of house prices across regions in China, this paper utilizes the FAVAR model to investigate the importance of national fundamental factors to the movement of regional house prices through common factors. The empirical results are as follows.

Firstly, our impulse response analysis shows that monetary shocks (R and M2) can significantly affect regional house prices, but differently across regions. The effects of monetary shocks (R and M2) are very strong in some regions but very weak in other regions. This confirms the inference that the analysis only based on the national average of house prices may be misleading.

Secondly, we find that the effects of the interest rate and M2 are different. The interest rate shocks can mainly affect the house prices in the developed regions with high growth rates of house prices, whereas the M2 shocks can mainly affect the house prices in the regions with close geographical and economic proximities.

Thirdly, our impulse response analysis shows little evidence of the effects of the real output and inflation rate on the movement of regional house prices. This indicates that the real output and inflation rate may not be the main driving force for the house price boom in China.

Our study contributes to the literature in several aspects. Firstly, differently from others, as it examines the relationship from a regional perspective, we could identify the effects of national fundamentals on 30 regions' house prices by 30 impulse responses from a single FAVAR system with 154 variables; the high dimensional system may yield more statistically significant results. Secondly, the empirical results show that monetary shocks can significantly affect regional house prices through common factors, but the effects are pretty different across regions, which means that the monetary shocks can significantly affect regions with high growth rates instead of low growth rates. Thirdly, differently from others, we find that the effects of the national real output and inflation rate are less important to the 30 regions' house price movements.

Our empirical analysis has important implications for policymakers. The different effects of monetary shocks across regions imply that the policy formulation and evaluation would be misleading if it was only based on the national house price indicator. Moreover, our study offers valuable information for regulators to improve the effectiveness of monetary policy to stabilize house markets from a regional perspective; policymakers can design contractionary monetary policy to curb the overheating house prices in the regions with high growth rates of house prices.

**Author Contributions:** Conceptualization, X.G., W.K. and Z.H.; methodology, X.G.; software, X.G.; validation, X.G., W.K. and Z.H.; formal analysis, X.G.; investigation, X.G. and W.K.; resources, X.G.; data curation, X.G. and Z.H.; writing—original draft preparation, X.G.; writing—review and editing, X.G. and W.K.; visualization, X.G.; supervision, X.G.; project administration, X.G.; funding acquisition, X.G. All authors have read and agreed to the published version of the manuscript.

**Funding:** This research was funded by Humanities and Social Sciences project of universities in Jiangxi Province, grant number GL20213; This research was funded by Science and Technology Research Project of Education Department of Jiangxi Province, grant number GJJ190283; This research was funded by Key Laboratory of Data Science in Finance and Economics, Jiangxi University of Finance and Economics, Nanchang, Jiangxi 330013, China.

**Informed Consent Statement:** Not applicable.

**Data Availability Statement:** 3rd Party Data Restrictions apply to the availability of these data. Data was obtained from [Wind database] and are available [from the Wind/at https://www.wind.com.cn/] with the permission of [Wind database].

**Conflicts of Interest:** The authors declare no conflict of interest.

**Appendix A**

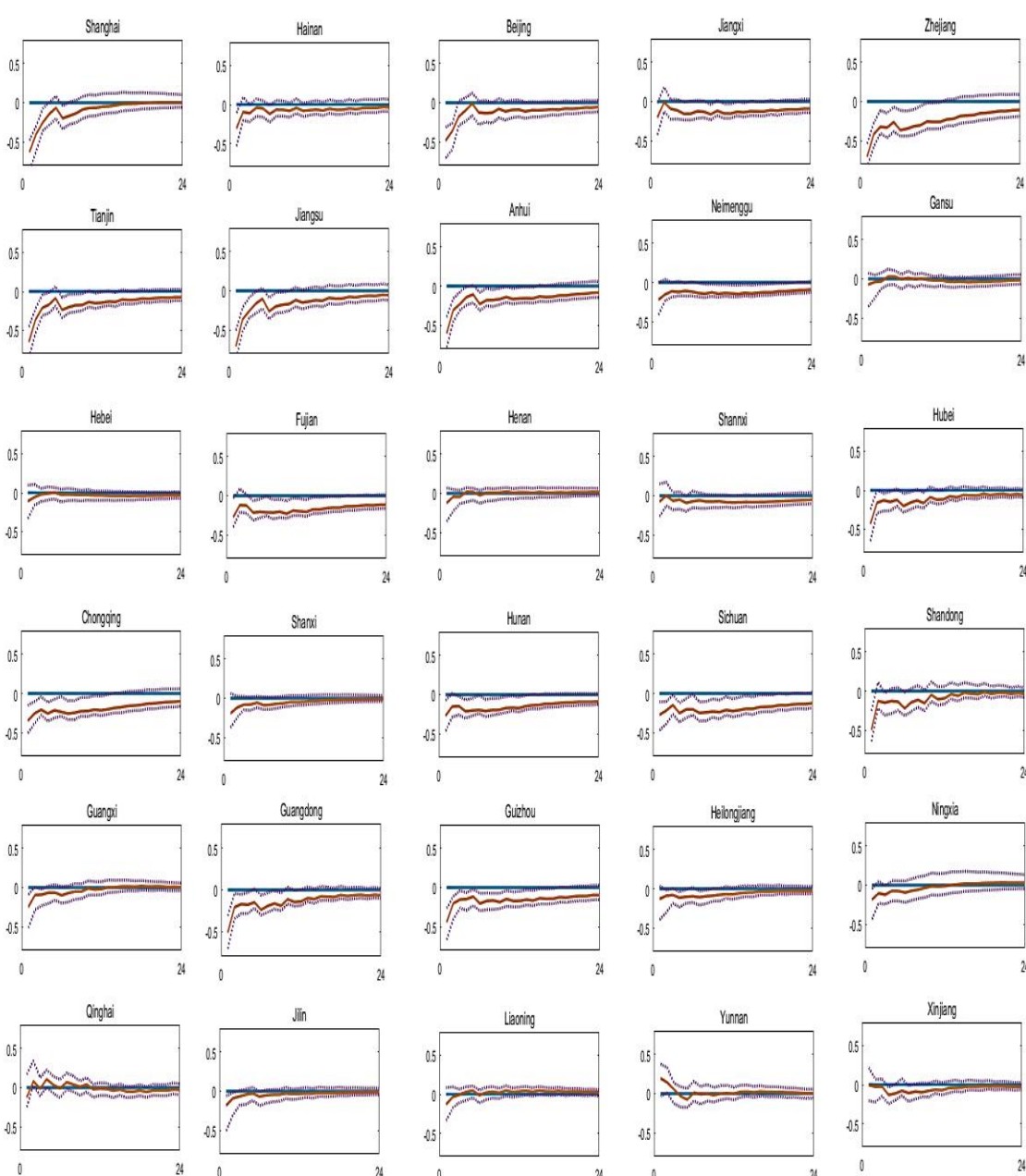

**Figure A1.** Impulse responses of regional house prices to one standard error shock to interest rate: FAVAR system with three factors.

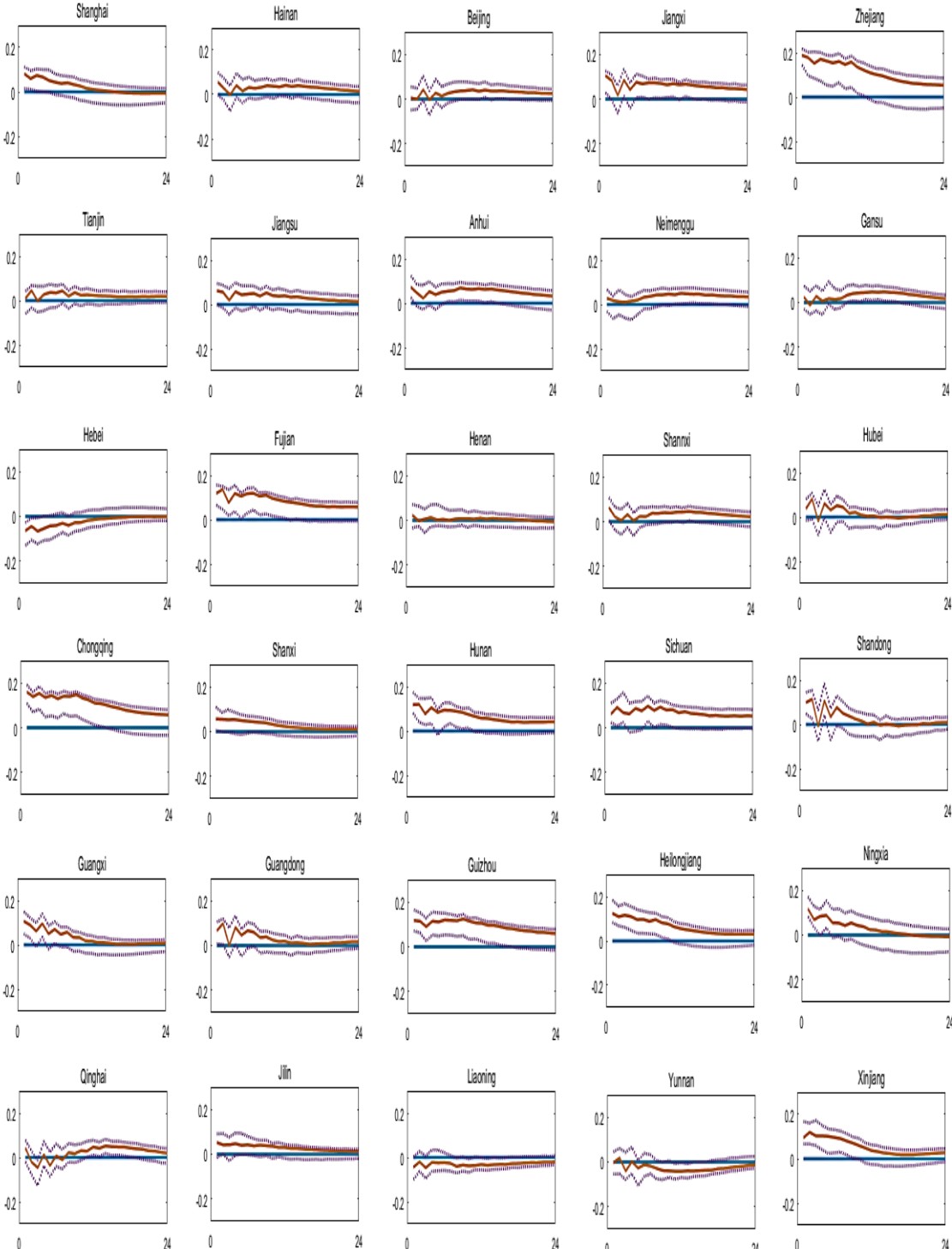

**Figure A2.** Impulse response of house prices to one standard error shock to M2: FAVAR system with three factors.

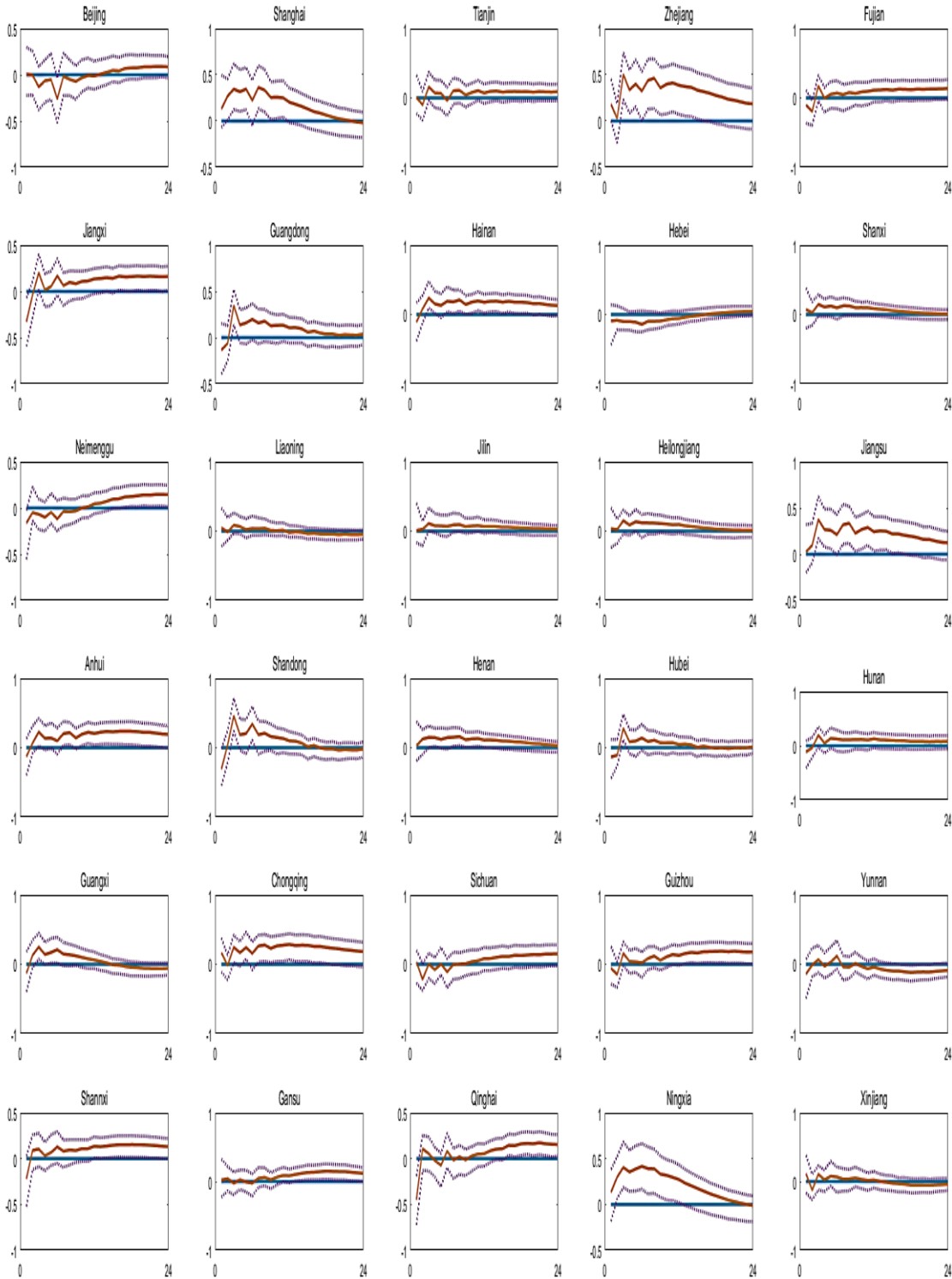

**Figure A3.** Impulse responses of regional house prices to one standard error shock to real output: FAVAR system with three factors.

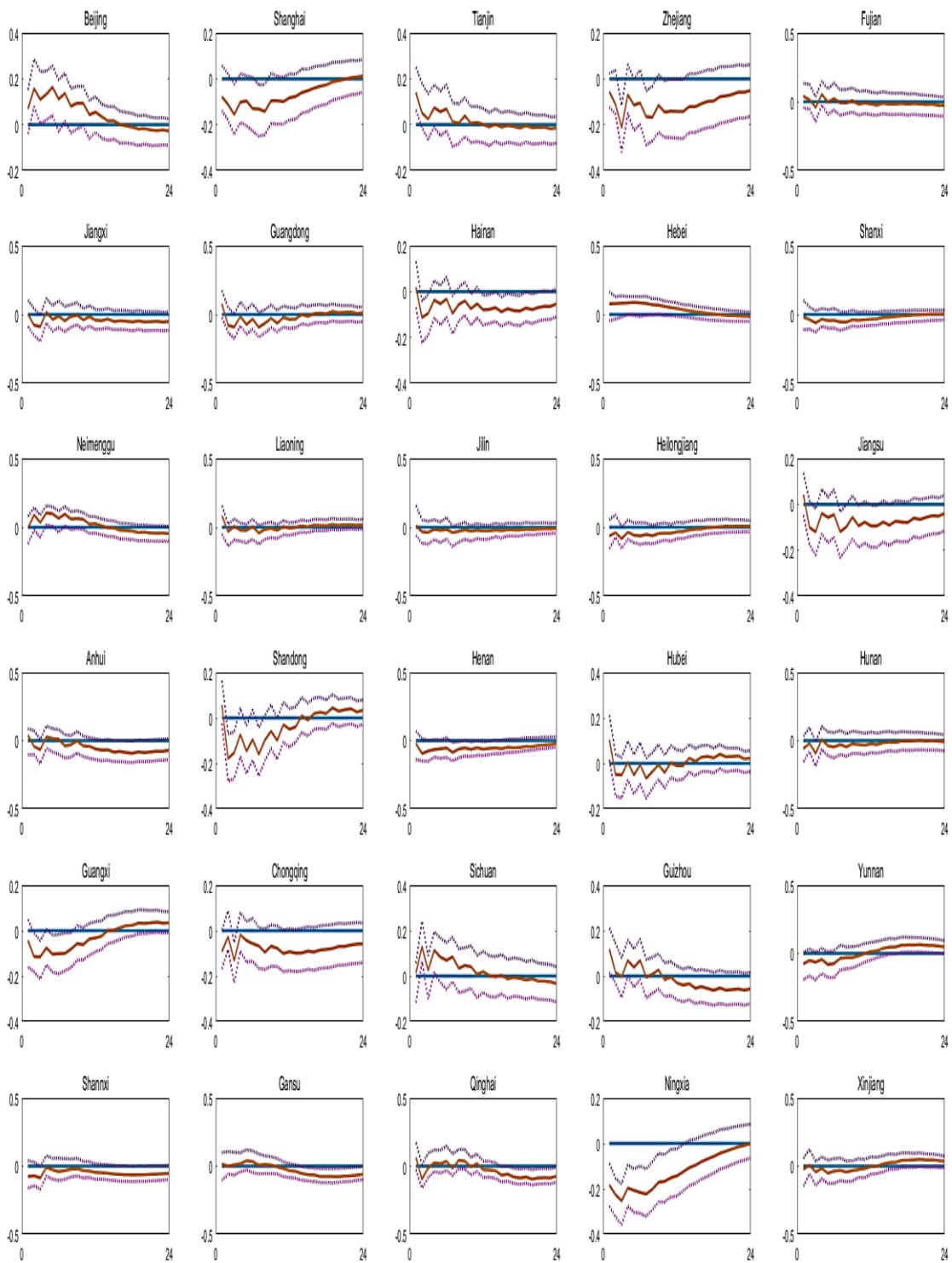

**Figure A4.** Impulse responses of regional house prices to one standard error shock to inflation rate: FAVAR system with three factors.

## Notes

[1]   In the case of the missing January data, we take the mean value of the previous month (December) and the next month (February). In the case of the missing February data, we take the mean value of the previous month (January) and the next month (March). In the case of the missing data from both months, firstly, we calculate the mean value of the next two months (March and April) to obtain the value of February; then, we take the mean values of December and February to obtain the value of January.

2  The Regional Coordinative Development Strategy is formulated on the Third Plenary Session of the 16th CPC Central Committee. It refers to actively promoting the development of the western region, revitalizing the old industrial bases such as the northeast region, promoting the rise of the central region, encouraging the eastern region to take the lead in development, continuing to give full play to the advantages and enthusiasm of each region, and gradually reversing the trend of widening the regional development gap by improving market mechanisms, cooperation mechanisms, mutual aid mechanisms, and support mechanisms so as to form a mutual promotion, complementary advantages, and a common ground between the east and the west. This is a new pattern of development.

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
