# Peer review of "The Effects of National Fundamental Factors on Regional House Prices: A Factor-Augmented VAR Analysis"

_jrfm, doi:10.3390/jrfm15070309_

Round 1

Reviewer 1 Report

The introduction must be rewrite. There are authors that are not in the references, there are references that are not in the main text. Please, argue the 7% annual growth level for dividing regions into two groups. Give more arguments for used indicators at national and regional level. Add in the abstract and in the conclusions the contributions of the paper.

Reviewer 2 Report

The topic of the paper is interesting and relevant, however, in its current form the manuscript is suitable for publishing. 

Major points:

1. The authors fail to convince me that the FAVAR methodology is suitable for answering the research question. I suggest describing the implementation of the FAVAR methodology in much more detail so that the reader is able to assess the merits of the methodology in this case and the results presented. 

E.g. how exactly are the 30 impulse responses for the 30 regions obtained? From a single 'FAVAR system' or from 30 individual FAVARs? In what sense does the FAVAR then reduce the dimension of the data? 

2. How exactly is the inference conducted, in particular how are the confidence intervals computed? 

3. The strength of the paper in my view is to document regional differences in the monetary policy impact on the housing market. Therefore, the nature of the analysis is first and foremost descriptive. Deriving far-reaching policy recommendations from such an analysis may be dangerous without more structure in the model.

4. The 'interpretation' of the results obtained and the 'explanations' provided are ex-post. They are not part of the model and therefore purely 'speculative stories'. I recommend being more cautious here. 

Take e.g. the remark: "Meanwhile, Guizhou and Ningxia are identified as the members of low growth group, the significant effects of interest rate on the house prices of these two regions may be attributed to the policy called as Regional Coordinative Development Strategy". This may be true or not. If you want to test for the effect of the policy measure discussed in Footnote 12, why not test for it explicitly?

In the same spirit:  If the authors want to study the effects of high growth regions as compared to other regions, the distinction should be made ex-ante between the two and be based on an exogenous variable. After all, why choose  7% as the threshold to divide the regions into two groups and not 6 or 8. E.g.

5. Related to my last remark. How are spatial effects taken care of in the analysis? At some stage, the authors mention spillover effects (however, again, as an ex-post 'interpretation' of the results).

6. The presentation of the impulse response results is quite confusing. Is there a more insightful way of presenting the results? Moreover, filling page after page with Figures of impulse responses based on a different number of factors does not add to the understanding. A simple comment may do and provides room for much-needed details on how the study is set up, as described above.

Minor points:

Chapter 2 closely follows the classical paper by Bernanke et al (2005) without any specific details for the study at hand. In particular, the discussion on the different estimation methods may not be necessary, when in the end the two-stage approach is employed in the study.

The content of Footnotes 8 and 10 are basically identical.

Why is it useful to provide 6 decimal places for the standard deviation of the IAV variable, when all the other variables carry 2 decimal places only?

Please carefully check for typos and grammatical errors.

Round 2

Reviewer 1 Report

Check again the text for spelling errors (page 6, last paragraph – 'moentary policy instruments').

Reviewer 2 Report

Regarding my comment no 1: I acknowledge the additional remarks introduced into the revised paper in order to provide more details. However, the additions are quite sparse and increase the readability of the paper only marginally.

Still an unanswered question in my view is the usefulness of reducing 150 variables to just a single factor. More information about the properties of this factor and the factor loadings would be helpful. 

Regarding my comment no 2: I do not agree with the answer provided. Typically authors using the two-step procedure recommend bootstrap methods to take into account the uncertainty in the factor estimation. See Stock and Watson (2002) Bai, Li and Lu (2016) and Yamamoto (2012).

Regarding my comment, no 4 and 5: referring to an additional paper that is not available at this stage and simply deleting respective remarks from the current manuscript is not very helpful in strengthening the content of the current paper as well as the confidence in its scientific contribution. 

Regarding my comment no 6: the 'solution' proposed is unsatisfactory. 

Minor comments:

I only noticed in the revised paper that the letter Y is used twice as a symbol in a different meaning. I suggest altering this.

on p. 6 remark on the Wind data base occurs twice.
